# OpenReview forum: "BLUR: A Benchmark for LLM Unlearning Robust to Forget-Retain Overlap"
_ICLR.cc/2026/Conference — Submitted to ICLR 2026_

### Official Review · Reviewer_oYmo · 2025-10-31

**Soundness:** 2
**Presentation:** 3
**Contribution:** 2
**Rating:** 2
**Confidence:** 5

**Summary:**

This paper addresses an important and practical challenge in LLM unlearning — how to properly evaluate and improve model behavior when there is overlap between the retention and unlearn datasets. The authors investigate the performance of existing unlearning methods under this overlapping condition, which often occurs in real-world applications but is rarely discussed in prior work.

On the newly proposed BLUR benchmark, existing unlearning methods exhibit clear signs of overfitting to the specific forgetting and retention targets within the current datasets. As a result, the proposed approaches underperform even the vanilla GA baseline, suggesting limited generalization capability under the more challenging and realistic overlap setting.

While the paper presents several empirical results, it does not propose any concrete method to improve unlearning performance in this more challenging overlapping scenario. Moreover, no theoretical justification or analytical insight is provided to explain the observed behaviors. As a result, the contribution remains largely empirical — consisting mainly of data-level manipulations, and in some cases, modifications of existing benchmarks rather than the introduction of entirely new datasets. As a result, the paper’s practical contribution and overall novelty are quite limited.

**Strengths:**

1. The problem addressed in this paper is important and timely. In practice, there is often substantial knowledge overlap between the forget and retain datasets, and studying this phenomenon is both meaningful and relevant for advancing research on LLM unlearning.

2. In the BLUR benchmark, the manipulation of existing dataset samples is simple yet effective. It provides an intuitive and clear way to illustrate how different unlearning methods behave under varying degrees of knowledge overlap.

3. The paper is well written, logically structured, and easy to follow. The motivation and experimental setup are clearly presented and easy to understand.

**Weaknesses:**

1. Regarding the originality of the proposed benchmark, it is important to note that BLUR is constructed by manipulating data from existing benchmarks. As a benchmark-oriented paper, the lack of newly collected data substantially limits the overall contribution of this work. Existing datasets were not originally designed with knowledge overlap in mind, and thus lack the necessary structure or optimization to support such analysis. Given that the proposed benchmark aims to specifically address the issue of overlap between forget and retain samples, it would be more convincing if the authors provided a dataset that was explicitly designed or curated for this purpose.

2. Considering the nature of overlap itself, there are essentially two distinct forms:

    (1) conceptual overlap, where the forget and retain QA pairs do not share identical questions but relate to the same underlying topic or keyword.

    (2) lexical overlap, where the forget and retain QA pairs are nearly identical, differing only by rephrasing.

These two scenarios may lead to very different behaviors in unlearning performance, and the paper would benefit from explicitly distinguishing and analyzing them.

3. From an experimental perspective, as a benchmark-focused paper, the comparisons across different methods are too limited. A broader range of existing and recent unlearning approaches [1-6] should be evaluated. Without extensive experimental validation or newly collected data, the work cannot be considered a strong benchmark contribution, as its impact remains insufficient.

At present, the paper falls into this category. However, if the authors were to include even a simple method that achieves a clear performance improvement under the new setup, the work could still make a meaningful contribution — positioning the benchmark as a discovery of a generally overlooked yet important challenge. Unfortunately, no such method is provided in the current version.

[1] Simplicity prevails: Rethinking negative preference optimization for llm unlearning NIPS2025

[2] Exploring Criteria of Loss Reweighting to Enhance LLM Unlearning. ICML2025

[3] Rethinking LLM Unlearning Objectives: A Gradient Perspective and Go Beyond. ICLR2025

[4] Llm unlearning via loss adjustment with only forget data. ICLR2025

[5] Your language model is secretly a reward model. NIPS 2024

[6]  Large language model unlearning via embedding-corrupted prompts. Arxiv 2406.07933

**Questions:**

Please refer to the weakness.

---

### Official Review · Reviewer_p2qi · 2025-10-31

**Soundness:** 2
**Presentation:** 3
**Contribution:** 2
**Rating:** 4
**Confidence:** 4

**Summary:**

Building on prior work, this paper proposes BLUR(a benchmark) and investigates two issues in machine unlearning: first, unlearning is more difficult when retained data overlaps with the data to be forgotten; second, after a simple relearning phase the model that had been unlearned often exhibits a rebound effect.

**Strengths:**

1. This paper addresses a practical problem of machine unlearning in LLMs.

2. Using four existing benchmarks, it evaluates MU methods comprehensively.

3. The writing is generally good and easy to understand.

**Weaknesses:**

1. The proposed benchmark appears to build heavily on prior work, as the authors acknowledge, which limits the paper's overall contribution. Moreover, the issue the authors describe, where retained data is related to the data to be forgotten, seems inherent to the definition of MU (i.e. what is the purpose of unlearning and which kinds of knowledge should be removed). The finding that relearning different amounts of forgotten data can reinstate the supposedly unlearned knowledge is unsurprising, since large language models tend to reproduce tokens and patterns that were frequent in their training data. Taken together, these points weaken the perceived novelty of the work.

2. There is a consensus that most existing unlearning methods, especially GA based approaches, suffer from unstable fine-tuning and are prone to model collapse. Some reported results show severe overfitting (for example, GA shows an extremely high perplexity in Table 6). This unfair comparison undermines the clarity of the paper.

3. As we all know that most MU methods exhibit a trade-off in forget quality and general utility, and all of the reported baselines shows this trend which makes it hard to distinguish which algorithm is better. This makes the claim "Contrary to recent work, we also see that simpler baselines such as gradient ascent often match or outperform more recent methods" less persuasive. In practice, you can simply fine-tune more steps to achieve a better forgetting and a worse knowledge retain. In addition, there is no explanation in the experiment section that justifies the chosen hyperparameters or demonstrates that the comparisons are fair across methods.

**Questions:**

1. Can you provide the full training configurations for each baseline? In addition, how do you ensure that the experimental comparisons are fair across methods (e.g., matched compute budget, identical hyperparameter search procedures, equivalent stopping criteria, and consistent random seeds)?

2. Please see weakness 1.

3. The tested baselines are training-based methods, how about those training-free methods? Do you think they still exist the same problem?

4. Do you think the problem addressed in your work (that relearning may recall unlearned knowledge) can be viewed as a general issue for modern LLMs, and would it also occur with other post‑training techniques such as model editing?

---

### Official Review · Reviewer_AndW · 2025-10-31

**Soundness:** 3
**Presentation:** 3
**Contribution:** 3
**Rating:** 2
**Confidence:** 4

**Summary:**

The paper proposes BLUR, a benchmark for evaluating Large Language Model (LLM) unlearning that is robust to forget-retain overlap.

The authors first identify critical weaknesses with existing unlearning benchmarks, noting that their "forget" and "retain" sets are highly disparate. This, they argue, paints a false and overly optimistic picture of unlearning effectiveness, as models remain vulnerable to "relearning attacks." They then propose the BLUR benchmark, which introduces more realistic and challenging evaluation scenarios. The benchmark includes 2 new kinds of evaluation dimensions:

(1) "Combined Queries," which semantically mix forget and retain information to test the model's ability to balance both, rather than evaluating them separately.

(2) "Relearning Data," a suite of benign datasets used to fine-tune the unlearned model to test if it "re-learns" the supposedly forgotten knowledge.

**Strengths:**

**Originality:** The paper significantly expands the concept of "forget-retain overlap" by defining three distinct forms of it. This provides a novel and broader perspective for evaluating LLM unlearning robustness.

**Significance:** This paper also provides three relearning datasets with different levels of relevance for the widely discussed topic of relearning attacks in the unlearning field.

**Quality:** The work's credibility is enhanced by building the new benchmark on top of existing, highly-regarded benchmarks (like TOFU, WMDP). This makes the findings more reliable.

**Weaknesses:**

**Limited Methodological Coverage:**

As a benchmark paper, the work has notable methodological limitations; it evaluates only 3 to 4 unlearning methods and fails to include several recent robust unlearning algorithms. This makes the benchmark less convincing as a comprehensive and representative evaluation.

**Lack of Deeper Insights:**

The paper lacks some novelty. Although it positions itself as a benchmark study, it should still provide deeper insights and high-level analyses for each newly introduced scenario, for example, explaining why different algorithms behave differently under the same setting. While the paper presents some “takeaways,” they read more like summaries of experimental phenomena rather than deep insights. For instance, in the first experiment (section 4.1), the authors only propose a brief hypothesis about why accuracy increases for some methods after insertion but drops for others, without offering any supporting visualization or quantitative evidence.

**Insufficient Experimental Details:**

(1) In the Verbatim Forget/Retain Combinations scenario, the authors place two questions within a single evaluation prompt, could this design itself introduce interference to the model’s behavior?

(2) In the Relearning experiments, the paper does not provide enough implementation details (e.g., number of fine-tuning epochs, training schedule, etc.), which are critical factors influencing relearning performance. As a benchmark paper, such omissions weaken the credibility of its results and analysis.

**Questions:**

See the questions in the “Weakness” section. This paper proposes some potential evaluation methods for future unlearning research, but I still have concerns about whether the design is truly useful or can offer meaningful insights for developing robust unlearning algorithms. I would consider increasing my score if the authors could address these concerns.

---

### Official Review · Reviewer_Rqmq · 2025-11-01

**Soundness:** 2
**Presentation:** 2
**Contribution:** 1
**Rating:** 2
**Confidence:** 4

**Summary:**

The paper introduces BLUR, a new benchmark for evaluating the robustness of LLM unlearning methods when forget and retain knowledge overlap. Existing unlearning benchmarks, such as TOFU, WMDP, and Who’s Harry Potter (WHP), assume clearly separated forget and retain datasets, which can lead to an overestimation of unlearning effectiveness. BLUR aims to provide a more realistic and challenging evaluation setting by introducing: Combined queries (keyword insertion, verbatim concatenation, and semantic overlap) and relearning benchmarks. The paper evaluates three unlearning baselines, including Gradient Ascent, Negative Preference Optimization, SCRUB, and their KL-regularized variants. Experiment results highlight that the tested unlearning methods are not robust to realistic evaluation conditions involving forget-retain overlap or benign retraining.

**Strengths:**

+ The paper studies an important and practical robustness problem in LLM unlearning.
+ BLUR introduces both combined query and relearning evaluation settings, capturing diverse forms of forget-retain overlap (keyword insertion, concatenation, and semantic) and multiple levels of relearning difficulty.
+ The BLUR is applied to well-known unlearning datasets (TOFU, WMDP, WHP, RWKU).

**Weaknesses:**

## Major Issues
+  Prior work [1] has already explored forget-token insertion into retain queries; this paper's extensions, e.g., multiple keyword replacement, combined queries, and relearning with varying relevance, build directly on this without introducing new methods or insights, making its originality limited. The proposed BLUR primarily extends existing datasets (TOFU, WMDP, WHP, RWKU) rather than introducing a new data generation pipeline or novel evaluation. As a result, the contribution is incremental.
+ BLUR is intended to serve as a robust evaluation benchmark, but important (robust) unlearning baselines are not evaluated, e.g., adversarially robust unlearning. This omission makes it difficult to assess the BLUR's practicality.
+ I carefully read the most relevant work to this paper [1]. In my assessment, the main findings of this study are not sufficiently novel and appear similar to or already discussed in prior work. For example, Takeaway 1 in [1] states that "Models that appear to have unlearned information when asked about the forget data and retain set separately may struggle on queries that ask about both forget and retain data." vs. Takeaway #2 in this study, "Popular unlearning heuristics achieve poor retain performance when the query contains both
forget and retain information."
+ Missing experimental details: no unlearning methods' formulation is described, no implementation details (hyperparameters, learning rate, epochs, coefficients, etc.) are included.
+ Figure 1 is the result of what datasets? Figure 1 shows that all methods are susceptible to combined queries, i.e., retain queries contain forget-tokens, but the results reported in Table 5 show that several unlearning methods (GA, NPO, SCRUB) are "robust" to keyword insertions, sometimes even showing improved accuracy. The reported results seem inconsistent.
+ It would be nice if the paper states clearly the problem formulation and threat model underlying BLUR's assumptions.
## Minor Issues
+ Typos: line 265/266 "Table 6" should be “Table 2”; line 151/152, 300/301 $w_u = \mathcal{M}(w,D_u)$ should be $w_u = \mathcal{M}_u(w,D_u)$.

## References

[1] Thaker, Pratiksha, et al. "Position: Llm unlearning benchmarks are weak measures of progress." 2025 IEEE Conference on Secure and Trustworthy Machine Learning (SaTML). IEEE, 2025.

**Questions:**

Please see the Weaknesses.

---

### Meta-Review · Area_Chair_tK6Z · 2026-01-03

**Summary:**

This paper proposes BLUR, a benchmark for evaluating LLM unlearning under forget-retain overlap, but reviewers broadly agree that its contribution is incremental and insufficiently substantiated as a standalone benchmark paper.

Across reviews, the dominant concern is limited novelty: BLUR is largely constructed by manipulating and extending existing benchmarks (TOFU, WMDP, WHP, RWKU), and many of its key observations closely mirror prior work that has already highlighted weaknesses of current unlearning evaluations under forget–retain overlap. As a benchmark paper, it also lacks breadth and representativeness, evaluating only a small set of unlearning methods while omitting several recent or more robust baselines, which undermines its practical value. Reviewers further point out insufficient experimental clarity and rigor, including missing details on hyperparameters, training schedules, threat models, and fairness of comparisons, as well as inconsistencies in reported results (e.g., between figures and tables). Finally, while the problem is important, the paper provides limited analytical insight beyond empirical observations and does not introduce new datasets or methods that clearly advance the state of the art.

The authors did not provide a rebuttal to address these concerns. Given the consensus on limited originality, incomplete evaluation, and lack of clarity, I recommend reject.

**Reviewer Concerns:**

The authors did not provide a rebuttal, so none of the reviewers’ concerns were substantively addressed.

**Reviewer Scores:**

The authors did not provide a rebuttal. As a result, none of the scores may be changed.

---

### Decision · Program_Chairs · 2026-01-26

Reject